# A Multispectral and Panchromatic Images Fusion Method Based on Weighted Mean Curvature Filter Decomposition

**Yuetao Pan** [1] , **Danfeng Liu** [1,*], **Liguo Wang** [1], **Shishuai Xing** [1] **and Jón Atli Benediktsson** [2]

1    College of Information and Communication Engineering, Dalian Minzu University, Dalian 116600, China
2    Faculty of Electrical and Computer Engineering, University of Iceland, 107 Reykjavik, Iceland
*    Correspondence: liudanfeng@dlnu.edu.cn

**Abstract:** Since the hardware limitations of satellite sensors, the spatial resolution of multispectral (MS) images is still not consistent with the panchromatic (PAN) images. It is especially important to obtain the MS images with high spatial resolution in the field of remote sensing image fusion. In order to obtain the MS images with high spatial and spectral resolutions, a novel MS and PAN images fusion method based on weighted mean curvature filter (WMCF) decomposition is proposed in this paper. Firstly, a weighted local spatial frequency-based (WLSF) fusion method is utilized to fuse all the bands of a MS image to generate an intensity component *IC*. In accordance with an image matting model, *IC* is taken as the original $\alpha$ channel for spectral estimation to obtain a foreground and background images. Secondly, a PAN image is decomposed into a small-scale (*SS*), large-scale (*LS*) and basic images by weighted mean curvature filter (WMCF) and Gaussian filter (GF). The multi-scale morphological detail measure (MSMDM) value is used as the inputs of the Parameters Automatic Calculation Pulse Coupled Neural Network (PAC-PCNN) model. With the MSMDM-guided PAC-PCNN model, the basic image and *IC* are effectively fused. The fused image as well as the *LS* and *SS* images are linearly combined together to construct the last $\alpha$ channel. Finally, in accordance with an image matting model, a foreground image, a background image and the last $\alpha$ channel are reconstructed to acquire the final fused image. The experimental results on four image pairs show that the proposed method achieves superior results in terms of subjective and objective evaluations. In particular, the proposed method can fuse MS and PAN images with different spatial and spectral resolutions in a higher operational efficiency, which is an effective means to obtain higher spatial and spectral resolution images.

**Keywords:** weighted mean curvature filter (WMCF); image matting model; multi-scale morphological measure (MSMDM); parameters automatic calculation pulse coupled neural network (PAC-PCNN)





## 1. Introduction

Nowadays, many remote sensing images with different resolutions can be obtained. The panchromatic (PAN) images have high spatial resolution and reflect the spatial structure information contained in the target region. The multispectral (MS) images contain rich spectral information, which is suitable for recognition and interpretation in the target region, but the spatial resolution is low. By fusing the MS and PAN images, the fused images have both higher spatial detail representation ability and retain the spectral features contained in the MS images, which can obtain a richer description about the target information. The fusion between the MS and PAN images is also known as pan-sharpening. With the improvement of remote sensing satellite sensor technology, the information integration and processing technology has been enhanced, and the pan-sharpening technology has been widely applied in various aspects such as military reconnaissance, remote sensing measurement, forest protection, mineral detection, image classification, and computer vision.

The pan-sharpening methods are divided into various fusion strategies. The component substitution-based (CS) methods mainly include the intensity hue saturation (IHS)

transform-based method [1], principal component analysis-based (PCA) method [2], adaptive Gram–Schmidt-based (AGS) method [3], etc. Moreover, Choi et al. [4] proposed a partial replacement-based adaptive component substitution-based (PRACS) method. This method generates high-resolution fused components by partial substitution and performs high-frequency injection based on statistical ratios. Vivone et al. [5] proposed a band-dependent spatial-detail with physical constrains-based (BDSD-PC) method. This method adds physical constraints to the optimization problem for guiding the band-dependent spatial-detail method toward a more robust solution. In general, the CS-based methods can effectively enhance the spatial resolution of the fused image. On the other hand, it may cause the spectral distortion to some extent.

The multi-resolution analysis-based (MRA) methods mainly include the Wavelet Transform-based (WT) [6], Curvelet Transform-based [7], and Non-Subsampled Shearlet Transform-based (NSST) methods [8]. With the MRA-based methods, the source image is disintegrated into several sub-images with different scales through some multi-scale decomposition methods. Then, different fusion rules are developed for sub-images at different scales, so that the sub-images can be fused. Finally, the final fused image is obtained by inverse transformation. In general, the MRA-based methods can obtain higher spectral resolution, but their spatial resolution is inferior in comparison with the CS-based methods.

Moreover, Fu et al. [9] proposed a variational local gradient constraints-based (VLGC) method. This method first calculates the gradient differences between the PAN and MS images in different regions and bands. Then, a local gradient constraint is added to the optimization objective, so as to fully utilize the spatial information contained in the PAN images. Wu et al. [10] proposed a multi-objective decision-based (MOD) method. This method designs a fusion model based on multi-objective decision making, which, in turn, performs a generalized sharpening operation by spectral modulation. Khan et al. [11] proposed a Brovey Transform-based method that integrates Brovey and Laplacian filter to improve the pan-sharpening effects.

In recent years, edge-preserving filters (EPF) are extensively applied for image processing. Li et al. [12] proposed a novel method in accordance with a guided filter. This method applies a guided filter to enhance the spectral resolution of the fused image. EPF can fully utilize the spatial information contained in the original images by preserving the edge information while smoothing it. Thus, EPF is often used as a decomposition method. Another important method is the Pulse Coupled Neural Network (PCNN) model [13]. After several iterations, the PCNN model can accurately extract the features contained in the image. Thus, it is suitable for the field of image fusion [14,15]. For the traditional PCNN model, the primary obstacle it faces is how to set the free parameters scientifically. Thus, the Parameters Automatic Calculation Pulse Coupled Neural Network (PAC-PCNN) model [16] is introduced into the pan-sharpening process. All parameters in the PAC-PCNN model can be automatically calculated according to the inputs with a fast convergence speed.

In general, the fusion images obtained by the CS-based methods can obtain higher spatial resolution, but many spectral information will be lost. The fusion images obtained by the MRA-based method can retain the spectral information, but will lose many spatial details. In particular, the fusion images obtained by the MRA-based methods tend to have spatial distortion, while the fusion images obtained by the CS-based method tend to have spectral distortion. It is important to balance spatial distortion and spectral distortion. Thus, a novel MS and PAN images fusion method based on weighted mean curvature filter (WMCF) decomposition is proposed. The proposed method combines the advantages of EPF decomposition and PAC-PCNN model, and focuses on solving spatial and spectral distortion problems. Firstly, a weighted local spatial frequency-based method (WLSF) is used to fuse all the bands of the MS image to generate intensity component *IC*. According to an image matting model, *IC* is used as the original $\alpha$ channel for spectral estimation to obtain a foreground and background images. Secondly, a PAN image is decomposed into a

small-scale (*SS*) image, a large-scale (*LS*) image and a basic image by WMCF and GF. The multi-scale morphological detail measure (MSMDM) values are used as the inputs of the PAC-PCNN model. With the PAC-PCNN model guided by the MSMDM values, the basic image and *IC* are effectively fused by the PAC-PCNN model. The fused image, and the *LS* and *SS* images are linearly combined together to construct the last $\alpha$ channel. Finally, the last $\alpha$ channel, foreground and background images are reconstructed in accordance with an image matting model to acquire the final fused image.

The four primary contributions of this paper are listed below:

(1)　An image matting model is introduced to effectively enhance the spectral resolution of the fused image. The preservation of spectral information contained in the MS image is mainly achieved by the image matting model;

(2)　A MSMDM method is used as a spatial detail information measure within a local region. By using the multi-scale morphological gradient operator, the gradient information of an original image can be extracted at different scales. Moreover, summing the multiscale morphological gradients helps both to measure the clarity and to suppress noise within a local region;

(3)　A PAC-PCNN model is introduced in the fusion process. The MSMDM values are taken as the inputs to the PAC-PCNN model. All parameters in the PAC-PCNN model are calculated automatically in accordance with the inputs and the conversion speed is also fast;

(4)　WLSF method improves the calculation by adding the diagonal direction based on the original spatial frequency. In addition, based on the Euclidean distance, it is determined that the weighting factor for the row and column frequencies is $\sqrt{2}$;

(5)　A WMCF method is used to decompose image with multi-resolution, which has advantages including robustness in scale and contrast, fast computation, and edge protection.

The remainder of the paper is organized below. Section 2 introduces the related methods, including an image matting model, a MSMDM method, a PAC-PCNN model, a WLSF method, and a WMCF method. Section 3 describes the fusion process in detail. Section 4 performs comparison experiments on four image pairs and analyzes the experimental results. Section 5 presents the summary and some future works.

## 2. Related Methods

### 2.1. Image Matting Model

On the basis of an image matting model [17], an input image *D* can be separated into a foreground image *F* and a background image *B* by a linear synthesis model, i.e., the color of the *i*-th pixel is a linear combination of the corresponding foreground color $F_i$ and background color $B_i$, as shown below:

$$D_i = \alpha_i F_i + (1 - \alpha_i) B_i \tag{1}$$

where $F_{ix}^j$ is the foreground color of the *i*-th pixel. $B_i$ is the background color of the *i*-th pixel. $\alpha$ is the opacity of the foreground image *F*. After the inputs *D* and original $\alpha$ channel are determined, the foreground image *F* and the background image *B* are evaluated through addressing the formula:

$$\min \sum_i \sum_j \left( \alpha_i F_i^j + (1 - \alpha_i) B_i^j \right)^2 + |\alpha_{ix}| \left( (F_{ix}^j)^2 + (B_{ix}^j)^2 \right) + |\alpha_{iy}| \left( (F_{iy}^j)^2 + (B_{iy}^j)^2 \right) \tag{2}$$

where *i* is the *i*-th channel. $F_{ix}^j$, $F_{iy}^j$, $B_{ix}^j$, $B_{iy}^j$, $\alpha_{ix}$, and $\alpha_{iy}$ are the horizontal and vertical derivatives of the foreground color $F^j$, background color $B^j$, and $\alpha$ channel, respectively.

### 2.2. Multi-Scale Morphological Detail Measure

A multi-scale morphological detail measure (MSMDM) method [18] is used as a spatial detail information measure within a local region. By the multi-scale morphological gradient operator, the gradient information of an original image can be extracted at different scales. Moreover, summing the multiscale morphological gradients of a local region helps both to measure the clarity and to suppress noise within a local region. The detailed implementation process of MSMDM is as follows:

Firstly, the multi-scale structural elements should be constructed, as detailed below:

$$TE_j = \underbrace{TE_1 \oplus TE_1 \cdots \oplus TE_1}_{t}, t \in \{1, 2, 3, \cdots, n\} \tag{3}$$

where $TE_1$ is a basic structure element whose radius is $r$, and $n$ is the number of scales.

The structural elements with different shapes can extract different kinds of features contained in the original image. Moreover, it can be extended to several scales by altering the size of the structural elements. Then, using the structure element, a comprehensive gradient feature can be extracted from the original image.

Then, a multi-scale morphological gradient operator is used to extract the gradient features $G_k$ at scale $t$ from an image $g$.

$$G_k(x, y) = g(x, y) \oplus TE_k - g(x, y) \ominus TE_k, k \in \{1, 2, 3, \cdots, n\} \tag{4}$$

where $\oplus$ and $\ominus$ denote the morphological expansion and corrosion operators, respectively.

The gradients can be expressed as local pixel value difference information in the original image. In particular, the maximum and minimum pixel values in the local area for an original image can be obtained by the expansion and erosion operators, respectively. In essence, the morphological gradient is the difference between the results obtained by expansion and erosion operations. Thus, the local gradient information contained in an original image can be extracted completely by the morphological gradient. Moreover, the gradients can be extracted at different scales by using multi-scale structural elements.

Moreover, the gradients at all scales are integrated into the multi-scale morphological gradient (MSMG).

$$\text{MSMG}(x, y) = \sum_{k=1}^{n} [w_k \times G_k(x, y)] \tag{5}$$

where $w_k$ denotes the weight of the gradient at scale $k$:

$$w_k = \frac{1}{2 \times k + 1} \tag{6}$$

The weighted summation is an effective method for fusing multi-scale morphological features. Thus, we can first multiply the gradient values at each scale with appropriate weights to obtain the morphological gradient values at each scale. Then, the morphological gradient values of all scales are summed, and, thus, the multiscale morphological gradient values are calculated.

In this paper, we assign larger weights to the smaller scale gradients and smaller weights to the larger scale gradients. In particular, for the smoother parts of the image, the pixel values vary less and the corresponding gradient values are smaller. Thus, the gradient-weighted sum of different scales can reflect the spatial detail information.

Finally, the multi-scale morphological gradients in local region are summed to calculate the MSMDM values and the specific details are shown in Formula (9). Summing the multi-scale morphological gradients can help measure the clarity as well as suppress the noise within a local region. Two examples of MSMDM are shown in Figure 1.

$$\text{MSMDM}(x, y) = \sum_{(p,q)} \text{MSMG}(p, q), (p, q) \in B \tag{7}$$

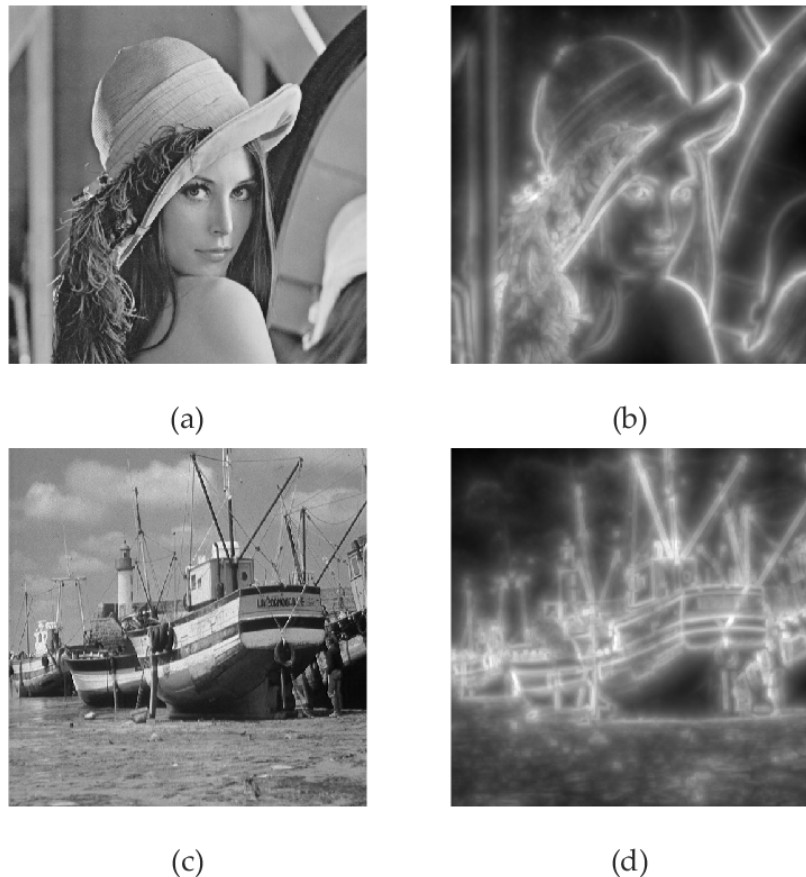

**Figure 1.** Two examples of MSMDM. (**a**) Input image of the first example; (**b**) MSMDM result of the first example; (**c**) Input image of the second example; and (**d**) MSMDM result of the second example.

There are three parameters to be set in MSMDM, i.e., the shape of the structural element, the size of the basic structural element and the number of scales. In this paper, a flat structure is chosen as the shape of the structure element. To suppress the noise, the radius of the basic structural element $TE_1$ is set to 4. Then the radius of the structural element at the $n$-th scale will be equal to $(2n + 1)$. Finally, the number $n$ of scales from 4 to 9 is tested experimentally, and the best results were obtained when $n = 6$.

### 2.3. Parameters Automatic Calculation Pulse Coupled Neural Network

The Pulse Coupled Neural Network (PCNN) model was proposed by Johnson et al. by improving and optimizing the Eckhorn model and Rybak model [13]. A single neuron consists of three parts: the input part, the link part, and the pulse generator. Compared with the artificial neural networks, the PCNN model does not require any training process. In the PCNN model, there is a one-to-one correspondence between the pixels of an image and neurons. The connection model of the PCNN neurons is shown in Figure 2.

For the traditional PCNN model, the primary obstacle it faces is how to set the free parameters scientifically. In order to overcome the challenges and set these parameters scientifically, a Parameters Automatic Calculation Pulse Coupled Neural Network (PAC-PCNN) model [16] is introduced into the pan-sharpening process. All parameters in the PAC-PCNN model are calculated automatically in accordance with the inputs and the conversion speed is also fast. In this paper, the MSMDM value of the original image is taken as the inputs of the PAC-PCNN model. Figure 3 demonstrates the structure of the PAC-PCNN model.

In the above PAC-PCNN model, $L_{ij}$ denotes the connection input at the position $(i, j)$. $S_{ij}$ denotes the input information at the position $(i, j)$. $V_L$ denotes the amplitude of the connection input and $U_{ij}$ denotes the internal activity item. $\alpha_f$ denotes the exponential

attenuation coefficient and $\beta$ denotes the connection strength. The output $Y_{ij}$ has two states: one is ignition ($Y_{ij} = 1$) and the other is non-ignition ($Y_{ij} = 0$). Its status depends on its two inputs, i.e., the current internal activity $U_{ij}$ and the previous dynamic threshold $E_{ij}$. Moreover, $\alpha_e$ and $V_E$ denote the exponential attenuation coefficient and the amplification coefficient of $E_{ij}$, respectively. $W$ denotes the following connection matrix, whose value is generally determined by experience.

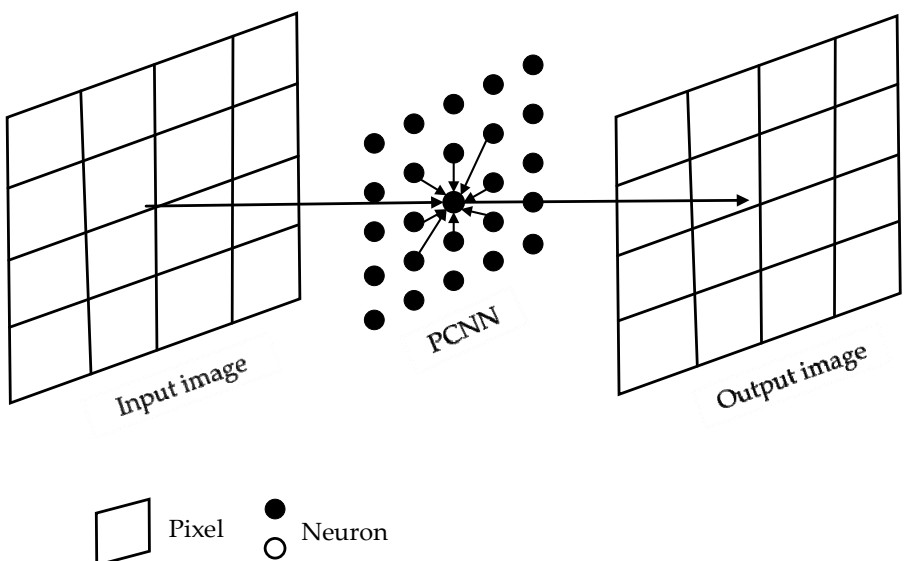

**Figure 2.** Connection model of the PCNN neurons.

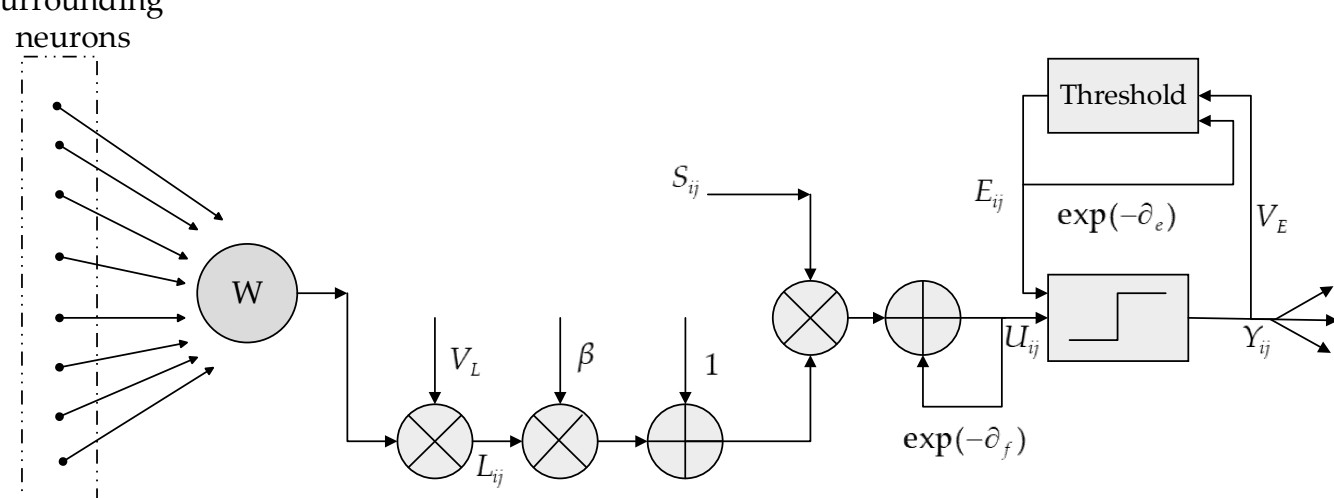

**Figure 3.** The structure of the PAC-PCNN model.

$$W = \begin{bmatrix} 0.5 & 1 & 0.5 \\ 1 & 0 & 1 \\ 0.5 & 1 & 0.5 \end{bmatrix} \tag{8}$$

The PAC-PCNN model is mainly used to segment images, but it is also an effective method to fuse images. In fact, the segmentation principle of PAC-PCNN-based image segmentation methods is basically based on pixel intensity. It means that the PAC-PCNN-based image fusion problem is strongly related to image segmentation. Thus, the PAC-PCNN model is introduced into the fusion process.

### 2.4. Weighted Local Spatial Frequency

The traditional spatial frequency (*SF*) [19] only describes spatial information in both horizontal and vertical directions and lack diagonal information, resulting in some texture and detail information is lost. However, this detail information is crucial for image fusion. *SF* is calculated by Formula (9).

$$SF(x,y) = \sqrt{RF^2(x,y) + CF^2(x,y)} \tag{9}$$

where $RF(x,y)$ and $CF(x,y)$ denote the row and column frequencies at the position $(x,y)$, respectively.

In this paper, the calculation of spatial frequency is improved by adding the calculation of diagonal direction on the basis of the original one. In addition, based on the Euclidean distance, it is determined that the weighting factor for row and column frequencies is $\sqrt{2}$. The improved spatial frequency is called weighted local spatial frequency (WLSF), which contains both row, column and diagonal directions, and the spatial frequencies in the eight directions are weighted and summed. WLSF can reflect the activity of each pixel in the neighborhood, and a larger spatial frequency value indicates that more spatial detail information is contained within a local area. WLSF is calculated as follows:

$$\text{WLSF}(x,y) = \sqrt{2RF^2(x,y) + 2CF^2(x,y) + DF^2(x,y)} \tag{10}$$

where $RF(x,y)$, $CF(x,y)$, and $DF(x,y)$ denote the row, column, and diagonal frequencies at the position $(x,y)$, respectively. *RF* and *CF* contain horizontal and vertical directions. *DF* contains positive and negative diagonal directions. The detailed definitions are as follows:

$$RF = RF_1 + RF_2, CF = CF_1 + CF_2 \tag{11}$$

$$DF = DF_1 + DF_2 + DF_3 + DF_4 \tag{12}$$

$$RF_1 = \sqrt{\frac{1}{(2M+1)(2N+1)} \sum_{j=-M}^{M} \sum_{k=-N}^{N} [P(x+j,y+k) - P(x+j,y+k-1)]^2} \tag{13}$$

$$RF_2 = \sqrt{\frac{1}{(2M+1)(2N+1)} \sum_{j=-M}^{M} \sum_{k=-N}^{N} [P(x+j,y+k) - P(x+j,y+k+1)]^2} \tag{14}$$

$$CF_1 = \sqrt{\frac{1}{(2M+1)(2N+1)} \sum_{j=-M}^{M} \sum_{k=-N}^{N} [P(x+j,y+k) - P(x+j-1,y+k)]^2} \tag{15}$$

$$CF_2 = \sqrt{\frac{1}{(2M+1)(2N+1)} \sum_{j=-M}^{M} \sum_{k=-N}^{N} [P(x+j,y+k) - P(x+j+1,y+k)]^2} \tag{16}$$

$$DF_1 = \sqrt{\frac{1}{(2M+1)(2N+1)} \sum_{j=-M}^{M} \sum_{k=-N}^{N} [P(x+j,k+n) - P(x+j-1,y+k-1)]^2} \tag{17}$$

$$DF_2 = \sqrt{\frac{1}{(2M+1)(2N+1)} \sum_{j=-M}^{M} \sum_{k=-N}^{N} [P(x+j,y+k) - P(x+j+1,y+k+1)]^2} \tag{18}$$

$$DF_3 = \sqrt{\frac{1}{(2M+1)(2N+1)} \sum_{j=-M}^{M} \sum_{k=-N}^{N} [P(x+j,y+k) - P(x+j-1,y+k+1)]^2} \tag{19}$$

$$DF_4 = \sqrt{\frac{1}{(2M+1)(2N+1)} \sum_{j=-M}^{M} \sum_{k=-N}^{N} [P(x+j,y+k) - P(x+j+1,y+k-1)]^2} \tag{20}$$

where $(2M + 1)(2N + 1)$ denotes the size of the local area. $P(x, y)$ denotes the pixel value at the position $(x, y)$.

### 2.5. Weighted Mean Curvature

The weighted mean curvature (WMC) [20] has the advantages of sampling invariance, scale invariance and contrast invariance as well as computational efficiency. With a given image $U$, the WMC is calculated as follows:

$$H^w(U) = n\|\nabla U\|_2 H(U) = \|\nabla U\|_2 (\nabla \cdot \frac{\nabla U}{\|\nabla U\|_2}) \tag{21}$$

where $\nabla$ and $\nabla\cdot$ denote the gradient operation and the scattering operation, respectively. In particular, for a two-dimensional image, i.e., $n = 2$, Formula (21) can be simplified as follows:

$$H^w(U) = \Delta U - \frac{U_y^2 U_{yy} + 2 U_x U_y U_{xy} + U_x^2 U_{xx}}{U_x^2 + U_y^2} \tag{22}$$

where $\Delta$ denotes the isotropic Laplace operator, $U_x$ and $U_y$ denote the partial derivatives in the $x$ and $y$ direction. $U_{xx}$, $U_{yy}$, and $U_{xy}$ denote the corresponding second-order partial derivatives. The detailed derivation process can be found in [20]. WMC can be considered as the gradient weighted by the mean curvature (MC) and also as the mean curvature weighted by the gradient. The connection between WMC, the gradient and MC is shown in Figure 4.

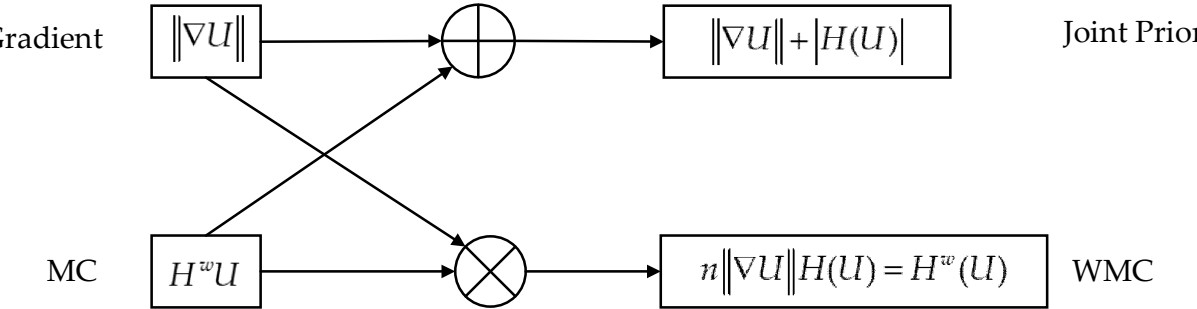

**Figure 4.** The connection between $\Delta U$, $H(U)$ and $H^w(U)$.

Formula (22) is a continuous form definition of WMC. However, the pixel points contained in the actual image are discrete. Thus, we should define a discrete form for WMC. For a $3 \times 3$ window, eight normal directions are considered. Figure 5 shows the eight possible normal directions.

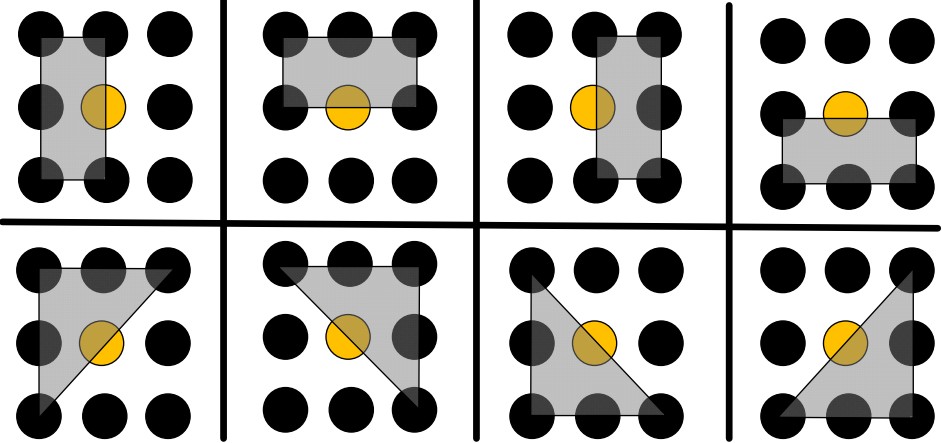

**Figure 5.** Eight possible normal directions.

The eight distances can be calculated from the above eight kernels as follows:

$$d_i = h_i * U, \forall i = 1, 2, \ldots, 8 \tag{23}$$

where $*$ denotes the convolution operation. The discrete form of Formula (22) is shown below:

$$H^w \approx d_k, k = \text{argmin}\{|d_i|; i = 1, 2, \ldots, 8\} \tag{24}$$

For convenience, the simplified process of WMC can be expressed as follows:

$$I_{out} = \text{WMC}(I_{in}) \tag{25}$$

where $I_{in}$ and $I_{out}$ are the input and filtered image, respectively. $\text{WMC}(\cdot)$ denotes the WMC filtering operation.

## 3. Methodology

### 3.1. Fusion Steps

Figure 6 shows the fusion steps of a novel MS and PAN images fusion method based on weighted mean curvature filter (WMCF) decomposition, and the detailed fusion process is described below:

(1)      Calculation the Intensity Component.

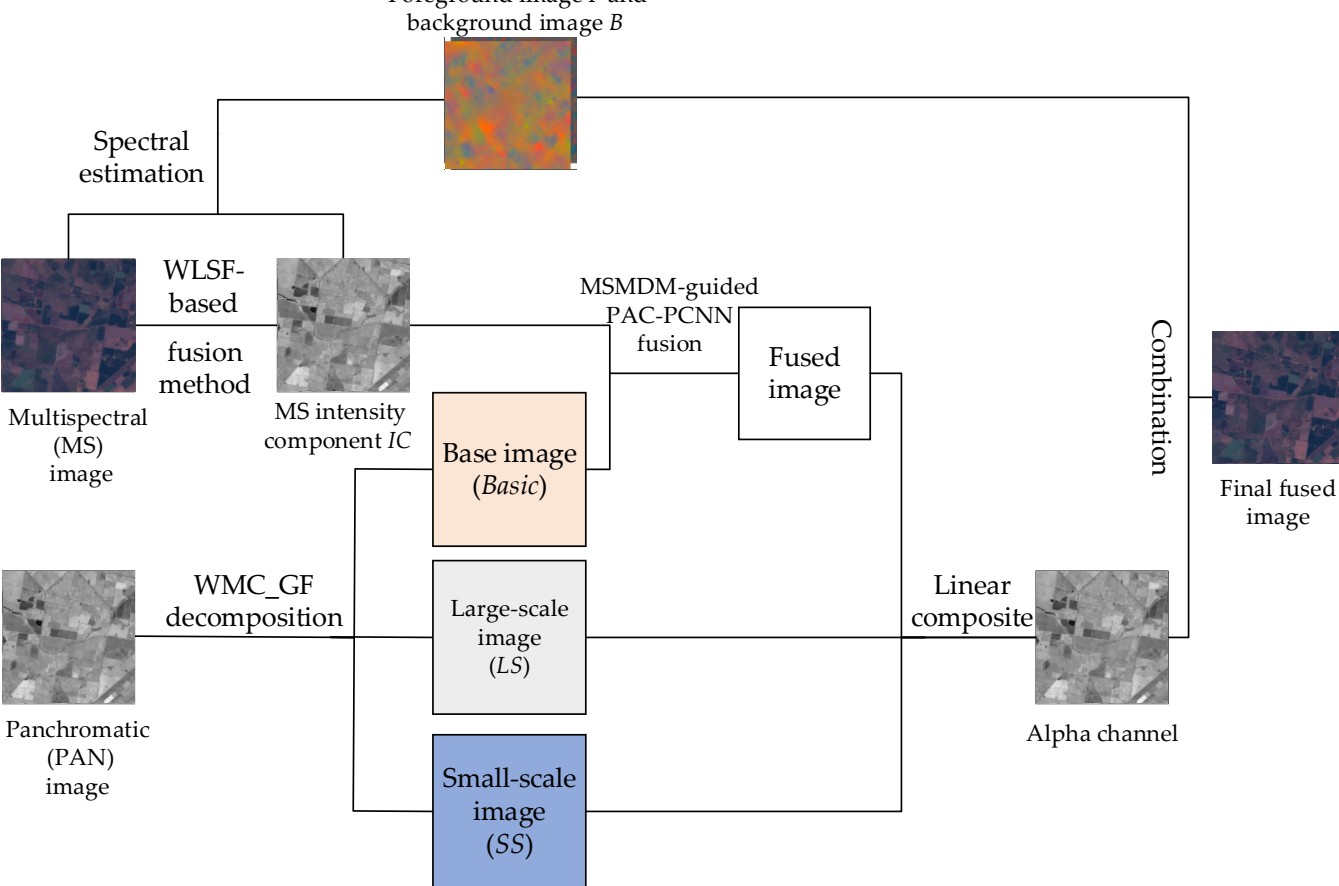

**Figure 6.** The fusion steps of the proposed method.

The main purpose of this step is to fuse all bands of the MS image in accordance with a WLSF-based fusion method, thus generating an intensity component.

If a simple averaging rule is used, some details and texture information contained in the original image will be dropped. Before performing the fusion operation, it is necessary

to weight the different regions in the image according to the importance of the information contained in the original image. The weighting factor size will directly affect the effect of fusion. Thus, a WLSF-based fusion method is utilized to fuse all the bands of a MS image to generate an intensity component *IC*. More specific details about WLSF can be found in Section 2.4.

An individual pixel fails to accurately represent all features within a local area. WLSF can make full use of the multiple pixels in a local area including horizontal, vertical, and eight neighborhoods in the primary and secondary diagonal directions. Different weights are assigned according to the Euclidean distance from the center pixel, and then participate in the weighted calculation. WLSF can be used as a quantitative index for important information such as details and textures. The pixels with larger WLSF values are more important for fusion and should be assigned with a larger weight, e.g., detail, texture, and other spatial detail information. Then, these pixels are given larger weights in the fusion process. Thus, we design a coefficient $\omega_d$ for adaptive weighted averaging based on the WLSF value, as shown below:

$$IC(i,j) = \sum_{d=1}^{n} \frac{1}{\omega_i(i,j)} MS_d(i,j) \tag{26}$$

$$\omega_d(i,j) = \frac{WLSF_d(i,j)}{\sum\limits_{d=1}^{n} WLSF_d} \tag{27}$$

where $n$ denotes the band number contained in the MS image. In the MS image, $WLSF_d(i,j)$ denotes the WLSF value of the $d$-th band at the position $(i,j)$, $MS_d(i,j)$ denotes the pixel value of the $d$-th band at the position $(i,j)$. $\omega_d(i,j)$ denotes the weighting factor of the $d$-th band at the position $(i,j)$. $IC(i,j)$ denotes the pixel value at the position $(i,j)$ in *IC*.

(2) Spectral Estimation.

The main purpose of this step is to extract the foreground and background colors in accordance with an image matting model, thus preserving the spectral information contained in the MS image.

Setting *IC* as the original $\alpha$ channel, a foreground image *F* and a background image *B* are obtained in accordance with Formula (2) in Section 2.1 to facilitate subsequent image reconstruction operations using *F* and *B*.

(3) Multi-scale Decomposition.

The main purpose of this step is to perform multi-scale decomposition for the PAN image on the basis of WMCF and Gaussian filter (GF), so as to sufficiently extract the spatial detail information contained in the PAN image.

The multi-scale decomposition is based on WMCF and GF. By Formulas (30)–(34) in Section 3.2, a PAN image can be decomposed into three sub-images with different scales, which are a large scale image *LS*, a small scale image *SS* and a basic image, respectively. More specific details about multi-scale decomposition can be found in Section 3.2.

(4) Component Fusion.

The main purpose of this step is to fuse the primary spatial information contained in the MS and PAN images in accordance with a MSMDM-guided PAC-PCNN fusion strategy, thus improving the spatial resolution of the final fused image.

The MSMDM value of the basic image and *IC* are used as the inputs of the PAC-PCNN model, respectively. All parameters in the PAC-PCNN model can be automatically calculated in accordance with the inputs with a fast convergence speed. In this paper, we set the maximum iteration number for the PAC-PCNN model to 2000. When the maximum number of iterations is reached, the iteration is stopped and then a fused image *FA* can be obtained. More specific details about component fusion can be found in Section 3.3.

(5)    Image Reconstruction.

The main purpose of this step is to substitute the final $\alpha$ channel, *F* and *B*, into an image matting model to obtain the final fused image.

A fused image *FB* is reconstructed through a linear combination of *SS*, *LS*, and *FA*, as shown in Formula (28). Then, *FB* is utilized as the last $\alpha$ channel. According to Formula (1) in Section 2.1, the final fusion result *HD* is calculated through combining the final $\alpha$ channel, *F*, and *B*, as shown in Formula (29).

$$FB = SS + LS + FA \tag{28}$$

$$HD = FB \times F + (1 - FB) \times B \tag{29}$$

Finally, the proposed method is summarized in the following Algorithm 1.

---

**Algorithm 1:** A WMCF-based pan-sharpening method

---

**Input:** low resolution MS image and high resolution PAN image
**Output:** high resolution MS image
1:    Calculation the Intensity Component
A WLSF-based fusion method is utilized to fuse all the bands of MS image to generate an intensity component *IC*.
2:    Spectral Estimation
Setting *IC* is the original $\alpha$ channel, a foreground image *F* and a background image *B* are obtained in accordance with Formula (2).
3:    Multi-scale Decomposition
Based on WMCF and GF, PAN is decomposed into three different scales by Formulas (30)–(34): large scale image *LS*, small scale image *SS* and basic image.
4:    Component Fusion
A MSMDM-guided PAC-PCNN fusion strategy is used to fuse the basic image and *IC*. Then, a fused image *FA* can be obtained.
5:    Image Reconstruction
A fused image *FB* is reconstructed through Formula (28). *FB* is utilized as the last $\alpha$ channel. According to Formula (1), the final fusion *HM* result is calculated through combining the final $\alpha$ channel, *F*, and *B*.
6:    **Return** *HM*

---

### 3.2. Multi-scale Decomposition Steps

WMCF can be used for image decomposition because it can preserve edge information while smoothing the image. In addition, Gaussian filter (GF) is a widely used image smoothing operator. In this paper, the multi-scale decomposition is based on WMCF and GF. The multi-scale decomposition of the PAN image using WMCF and GF obtains a large-scale image *LS*, a small-scale image *SS* and a basic image *Basic*. By this decomposition method, an input image can be decomposed into three sub-images with different scales. The specific decomposition process is as follows:

Firstly, the input image *M* is processed using WMCF and GF to obtain the filtered images $I_w$ and $I_g$, respectively.

Secondly, *SS* is obtained by the difference of *M* and $I_w$. *LS* is obtained by the difference of $I_w$ and $I_g$.

Finally, $I_g$ is used as the basic image, i.e., *Basic*. Formulas (30)–(34) show the specific computational procedure. The flow chart of the image decomposition method based on WMCF and GF is shown in Figure 7.

$$I_g = \text{GF}(M, \delta, \rho) \tag{30}$$

$$I_w = \text{WMCF}(M) \tag{31}$$

$$SS = M - I_w \tag{32}$$

$$LS = I_w - I_g \tag{33}$$

$$Basic = I_g \tag{34}$$

where $\rho$ and $\delta$ represent the size of the radius and the variance of the GF. $M$ represents a PAN image. WMCF$(\cdot)$ and GF$(\cdot)$ represent the GF and WMCF operation, respectively.

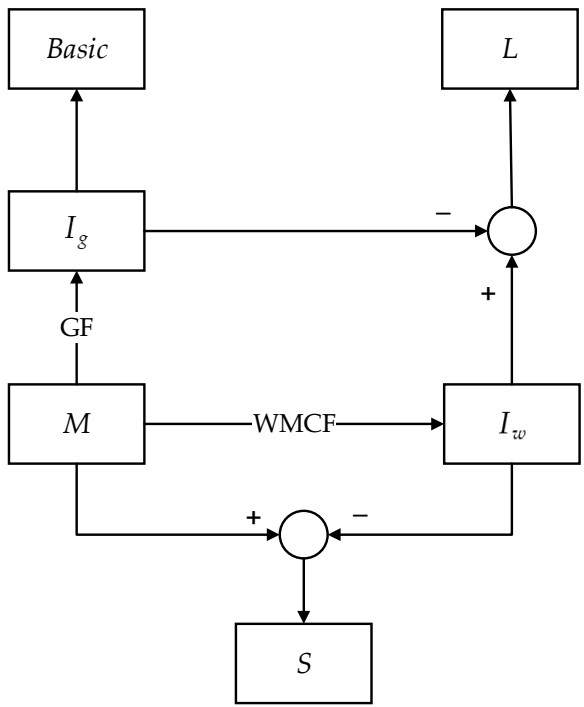

**Figure 7.** The flow chart of the multi-scale decomposition method based on WMCF and GF.

*3.3. Component Fusion Steps*

The PAC-PCNN model is introduced into the image fusion process. The MSMDM values of the basic image and MS intensity component *IC*, respectively, are used as the inputs of the PAC-PCNN model. The detailed design of the MSMDM is shown in Section 2.2.

There are five free parameters in the PAC-PCNN model: $\alpha_f, \beta, V_L, V_E, \alpha_e$. The weighted connection strength is denoted by $\lambda = \beta V_L$. Thus, the weighted connection strength is expressed. According to the analysis in [16], all the free parameters can be calculated adaptively according to the input information, which solves the difficulty of setting free parameters in the traditional PCNN model. All the free parameters in the PAC-PCNN model are automatically calculated in accordance with Formulas (35)–(38):

$$\alpha_f = \ln\left(\frac{1}{\sigma(S)}\right) \tag{35}$$

$$\lambda = \frac{\frac{S_{\max}}{S'} - 1}{6} \tag{36}$$

$$V_E = e^{-\alpha_f} + 1 + 6\lambda \tag{37}$$

$$\alpha_e = \ln\left(\frac{\frac{V_E}{S'}}{\frac{1-e^{-3\alpha_f}}{1-e^{-\alpha_f}} + 6e^{-\alpha_f}\lambda}\right) \tag{38}$$

where $\sigma(S)$ is the standard deviation of the normalized input image $S$. $S'$ and $S_{\max}$ indicate the normalized Ostu threshold and the maximum intensity of the input image, respectively.

When the maximum number of iterations is reached, the iteration is stopped. Then, the sum of the ignition times with the basic image and *IC* are obtained, respectively. That will provide the total number of ignitions $T_{Basic}$ and $T_{IC}$ for the basic image and *IC*, respectively.

The fusion results *Fused* are acquired through using the large number of ignition times. The fusion rules are as follows:

$$Fused(x,y) = \begin{cases} IC(x,y), T_{IC} \geq T_{Basic} \\ Basic(x,y), T_{IC} \leq T_{Basic} \end{cases} \quad (39)$$

where $Fused(x,y)$ represents the fused value at the position $(x,y)$ in the fusion image, $IC(x,y)$ represents the pixel value at the position $(x,y)$ in *IC*, and $Basic(x,y)$ represents the pixel value at the position $(x,y)$ in the basic image.

## 4. Experiments and Analysis

### 4.1. Datasets

We used a dataset which contains 36 image pairs [21]. Each image pair contains both MS and PAN images, and their pixel sizes are $200 \times 200$ and $400 \times 400$, respectively.

Firstly, we up-sampled the original MS image to obtain the MS image with pixel size of $400 \times 400$. Then, the MS image and PAN images are down-sampled to acquire the MS image and the PAN image with pixel size of $200 \times 200$ as the experimental images. In this case, the final obtained image is used as the experimental image, and the original image is used as the reference image.

We choose four image pairs from different scenes for comparison experiments. Figure 8 shows four image pairs containing MS and PAN images and will be utilized for the comparison experiments.

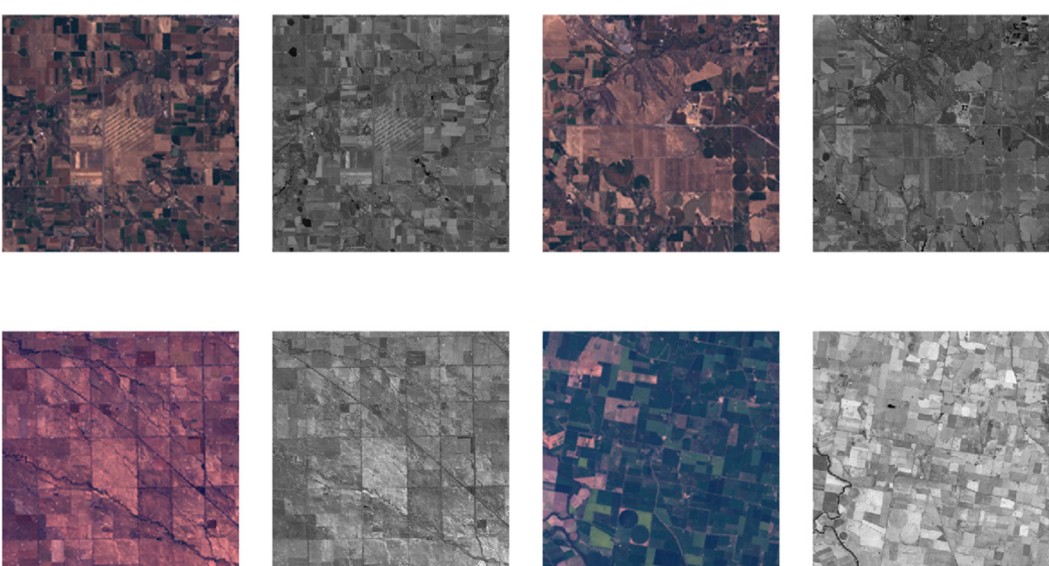

**Figure 8.** Four image pairs.

### 4.2. Comparison Methods

In this paper, ten existing pan-sharpening methods are used for comparison with the proposed method. These ten methods are BL [11], AGS [3], GFD [12], IHST [1], MOD [10], MPCA [2], PRACS [4], VLGC [9], BDSD-PC [5], and WTSR [6], respectively.

### 4.3. Objective Evaluation Indices

There are two ways to evaluate a pan-sharpening methods: subjective visual effects and objective evaluations. Each objective evaluation index considers different dimensions of the problem, and generally has an overall consistency, which can reflect the true image fusion results and also be consistent with the subjective evaluation indices. In order to objectively evaluate the quality of the fused images obtained by each method, five widely used quantitative indices are used in this paper. Some detailed introduction of each index is shown below:

(1) Correlation Coefficient (CC) [22]. It can calculate the correlation between the reference MS image and a fused image. Its optimum value is 1;

(2) Erreur Relative Global Adimensionnelle de Synthse (ERGAS) [23]. It can reflect the overall quality of a fused image and its optimum value is 0;

(3) Relative Average Spectral Error (RASE) [24]. It can reflect the average performance on spectral errors and its optimum value is 0;

(4) Spectral Information Divergence (SID) [25] can evaluate the divergence between spectra and its optimum value is 0. For more specific details about the SID index, please refer to the literature [25];

(5) No Reference Quality Evaluation (QNR) [26]. When without a reference MS image, it can reflect the overall quality of a fused image. QNR is composed of two parts: a spectral distortion index $D_\lambda$ and a spatial distortion index $D_s$. Its optimum value is 1. For QNR, a higher value indicates a better fusion effect.

$$QNR = (1 - D_\lambda)(1 - D_S) \tag{40}$$

(6) $D_\lambda$ can reflect the spectral distortion [26]. For $D_\lambda$, a lower value indicates a better fusion effect and its optimum value is 0;

(7) $D_s$ can reflect the spatial distortion [26]. For $D_s$, a lower value indicates a better fusion effect and its optimum value is 0.

### 4.4. Experimental Results and Analysis

In Figures 9–12, give four groups visualization results obtained by BL, AGS, GFD, IHST, MOD, MPCA, PRACS, VLGC, BDSD-PC, and WTSR, and the proposed method on the first image pair, the second image pair and the third image pair, respectively. Moreover, the last one gives the MS image as reference. In order to more visually compare the differences between the fusion results obtained by each method, all fusion results were locally enlarged.

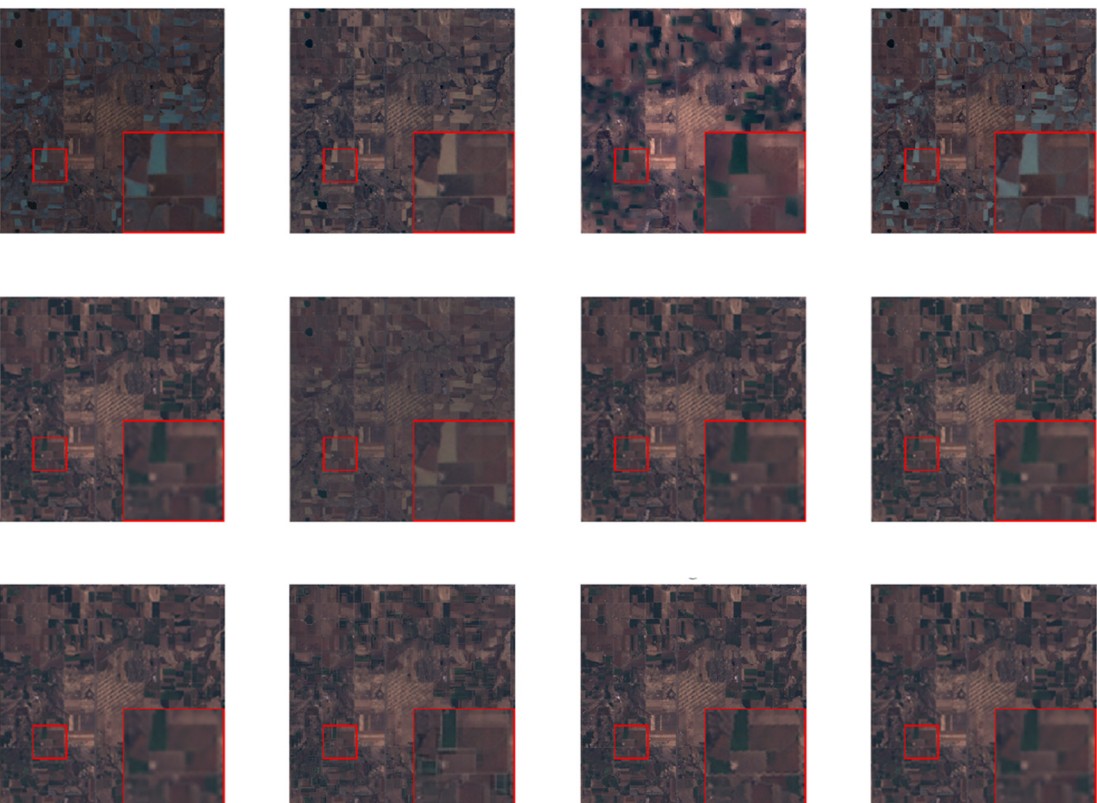

**Figure 9.** The first group of visualization results.

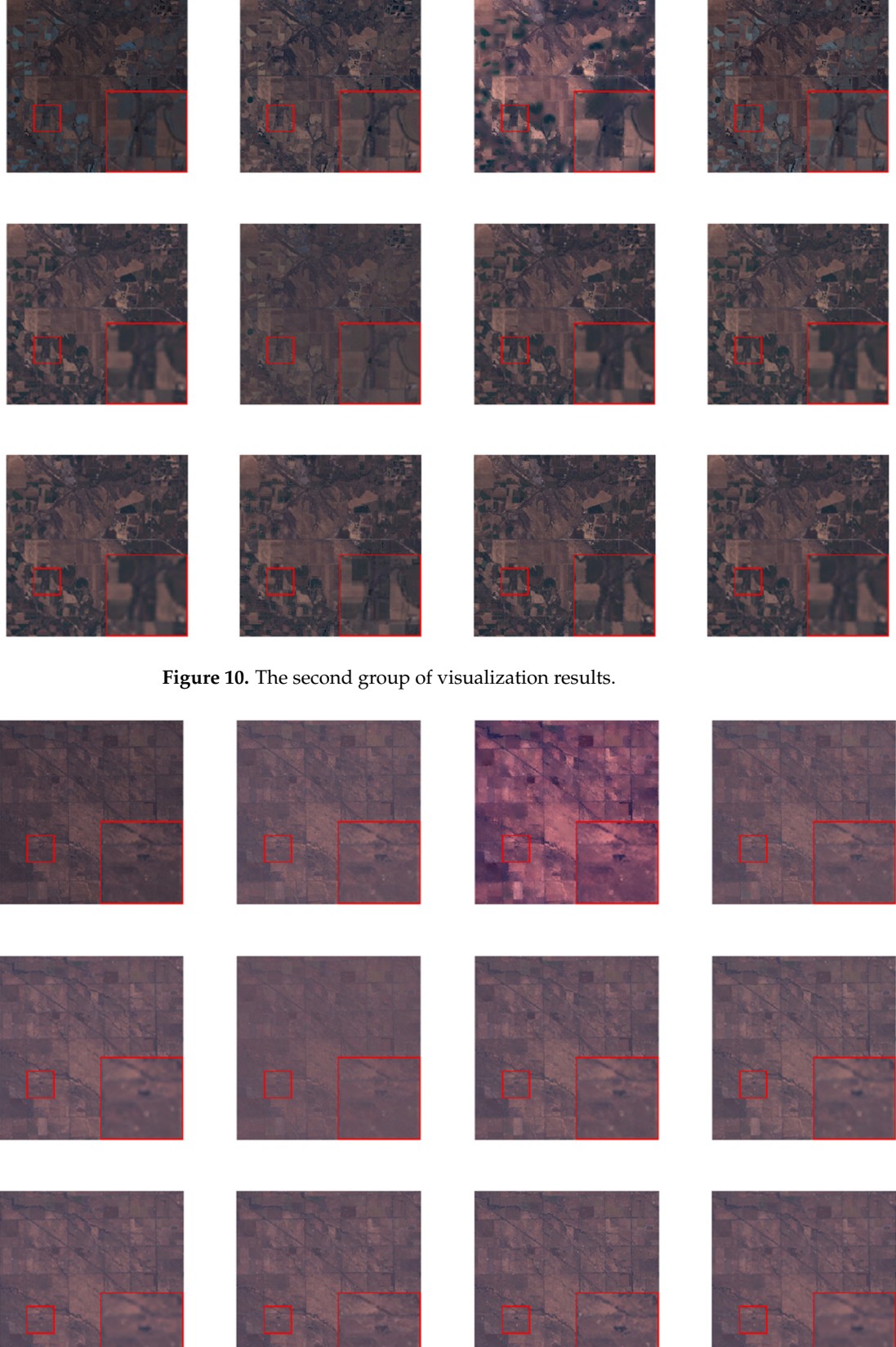

**Figure 10.** The second group of visualization results.

**Figure 11.** The third group of visualization results.

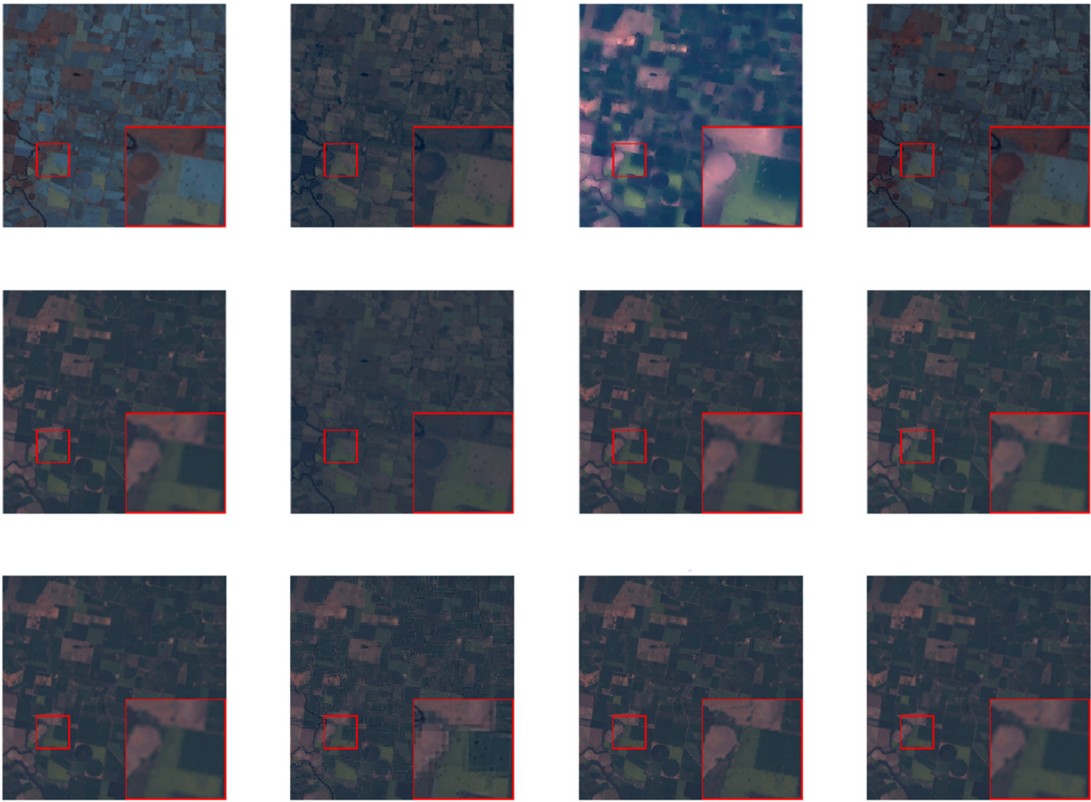

**Figure 12.** The fourth group of visualization results.

Tables 1–4 show four groups of quantitative results for the fusion results obtained by eleven different pan-sharpening methods on the four image pairs, respectively. There are five quantitative indices in total, including spectral and spatial quality evaluation. These five indices are Correlation Coefficient (CC), Erreur Relative Global Adimensionnelle de Synthse (ERGAS), Relative Average Spectral Error (RASE), Spectral Information Divergence (SID) and No Reference Quality Evaluation (QNR). In particular, the best values for all quantitative indices are displayed in bold red. The second best values for all quantitative indices are displayed in bold green. The third best values for all quantitative indices are displayed in bold blue.

**Table 1.** The first group of quantitative results.

|         | CC (1) | ERGAS (0) | SID (0) | RASE (0) | QNR (1) |
|---------|--------|-----------|---------|----------|---------|
| BL      | 0.663  | 7.519     | 0.014   | 20.753   | 0.586   |
| AGS     | 0.494  | 8.804     | 0.019   | 28.837   | 0.373   |
| GFD     | 0.755  | 7.216     | 0.010   | 16.435   | 0.629   |
| IHST    | 0.663  | 7.643     | 0.009   | 24.117   | 0.521   |
| MOD     | 0.945  | 1.801     | 0.004   | 7.334    | 0.887   |
| MPCA    | 0.500  | 7.900     | 0.011   | 24.146   | 0.552   |
| PRACS   | 0.943  | 1.813     | 0.008   | 7.369    | 0.784   |
| VLGC    | 0.946  | 1.789     | 0.004   | 7.358    | 0.863   |
| BDSD-PC | 0.945  | 1.809     | 0.003   | 7.335    | 0.843   |
| WTSR    | 0.841  | 3.044     | 0.006   | 12.392   | 0.498   |
| Proposed| 0.948  | 1.488     | 0.002   | 4.956    | 0.948   |

In Figure 9, compared with the reference MS image, the BL, AGS, IHST, and MPCA methods appear spectral distortion with different degrees in the overall region. In particular, for the BL method, the green part becomes dark blue in the local magnification region. For the GSA and MPCA methods, the green part becomes yellowish in the local magnification

area. Moreover, the WTSR and GFD method have some distortion with different degrees in the spatial details. The fused images obtained by the MOD, PRACS, VLGC, and BDSD-PC methods have higher spectral resolution and retain the spectral information contained in the MS images. However, compared with these methods, the spatial details of the proposed method are more defined, especially in the local magnification region. In Figure 7, from the analysis of subjective vision, the proposed method has clearer spatial details and higher spectral resolution, which means that the proposed method improves spatial details while more spectral information contained in the MS images is retained.

**Table 2.** The second group of quantitative results.

|          | CC (1) | ERGAS (0) | SID (0) | RASE (0) | QNR (1) |
|----------|--------|-----------|---------|----------|---------|
| BL       | 0.721  | 5.093     | 0.010   | 23.304   | 0.495   |
| AGS      | 0.491  | 5.986     | 0.018   | 28.299   | 0.316   |
| GFD      | 0.768  | 5.614     | 0.015   | 22.543   | 0.676   |
| IHST     | 0.735  | 5.509     | 0.009   | 21.980   | 0.550   |
| MOD      | 0.949  | 1.765     | 0.002   | 7.202    | 0.894   |
| MPCA     | 0.592  | 5.120     | 0.010   | 20.987   | 0.504   |
| PRACS    | 0.930  | 1.860     | 0.003   | 7.566    | 0.780   |
| VLGC     | 0.952  | 1.730     | 0.002   | 7.124    | 0.956   |
| BDSD-PC  | 0.949  | 1.765     | 0.002   | 7.200    | 0.884   |
| WTSR     | 0.869  | 2.764     | 0.006   | 11.273   | 0.567   |
| Proposed | 0.950  | 1.488     | 0.001   | 4.956    | 0.984   |

**Table 3.** The third group of quantitative results.

|          | CC (1) | ERGAS (0) | SID (0) | RASE (0) | QNR (1) |
|----------|--------|-----------|---------|----------|---------|
| BL       | 0.796  | 2.658     | 0.010   | 8.092    | 0.548   |
| AGS      | 0.793  | 3.697     | 0.016   | 9.058    | 0.435   |
| GFD      | 0.845  | 2.157     | 0.015   | 7.627    | 0.648   |
| IHST     | 0.731  | 1.674     | 0.009   | 6.741    | 0.515   |
| MOD      | 0.913  | 1.106     | 0.002   | 4.595    | 0.894   |
| MPCA     | 0.798  | 1.851     | 0.017   | 7.707    | 0.535   |
| PRACS    | 0.887  | 1.233     | 0.019   | 5.068    | 0.851   |
| VLGC     | 0.912  | 1.120     | 0.003   | 4.651    | 0.909   |
| BDSD-PC  | 0.911  | 1.106     | 0.006   | 4.595    | 0.884   |
| WTSR     | 0.836  | 1.448     | 0.013   | 5.966    | 0.640   |
| Proposed | 0.914  | 1.488     | 0.001   | 4.956    | 0.946   |

**Table 4.** The fourth group of quantitative results.

|          | CC (1) | ERGAS (0) | SID (0) | RASE (0) | QNR (1) |
|----------|--------|-----------|---------|----------|---------|
| BL       | 0.442  | 6.683     | 0.050   | 28.734   | 0.584   |
| AGS      | 0.313  | 7.504     | 0.065   | 26.254   | 0.351   |
| GFD      | 0.757  | 5.611     | 0.021   | 20.598   | 0.568   |
| IHST     | 0.464  | 6.129     | 0.029   | 26.129   | 0.597   |
| MOD      | 0.943  | 1.604     | 0.007   | 5.147    | 0.903   |
| MPCA     | 0.473  | 6.240     | 0.049   | 26.640   | 0.558   |
| PRACS    | 0.943  | 1.606     | 0.009   | 5.174    | 0.816   |
| VLGC     | 0.944  | 1.601     | 0.008   | 5.157    | 0.901   |
| BDSD-PC  | 0.942  | 1.611     | 0.007   | 5.159    | 0.884   |
| WTSR     | 0.733  | 3.053     | 0.021   | 11.237   | 0.573   |
| Proposed | 0.947  | 1.488     | 0.005   | 4.956    | 0.925   |

In Table 1, the proposed method can obtain the best values on all six quantitative indices. Moreover, the VLGC method can obtain the second best values on the CC and ERGAS indices, and the third best values on the SID and QNR indices. The MOD method can obtain the second best values on the RASE and QNR indices, and the third best values on the CC, ERGAS, and SID indices. The BDSD-PC method can obtain the second best

value on the SID index and the third best values on the CC and RASE indices. Thus, from the perspective of objective evaluation in Table 1, the proposed method has superior spatial detail retention characteristics and spectral retention characteristics, and the overall effect is better.

In Figure 10, compared with the reference MS image, the BL, AGS, IHST, and MPCA methods appear spectral distortion with different degrees in the overall region. Especially, in the local magnification region, for the BL method, the dark green part becomes dark blue. For the AGS and MPCA methods, the dark green part becomes yellowish. For the IHS method, the dark green part becomes light green. Moreover, the WTSR and GFD methods have severe spatial detail distortion. The fused images obtained by the MOD, PRACS, VLGC, and BDSD-PC methods have a higher spectral resolution and retain the spectral information contained in the MS images. However, compared with these methods, the spatial details of the proposed method are more defined, especially in the local magnification region. In Figure 8, from the analysis of subjective vision, the proposed method has clearer spatial details and higher spectral resolution, which means that the proposed method improves spatial details while more spectral information contained in the MS images is retained.

In Table 2, the proposed method can obtain the best values on all six quantitative indices. In particular, the QNR value of the proposed method is 0.984, which is relatively close to the optimal value of 1. In addition, the VLGC method can obtain the second best values on all six indices. The MOD method can obtain the second best value on the SID index and the third best values on the CC, ERGAS and QNR indices. The BDSD-PC method can obtain the second best value on the SID index and the third best values on the CC and RASE indices. The PRACS method can obtain the third best value on the SID index. Thus, from the analysis of the objective evaluation results in Table 2, the proposed method has superior spatial detail retention characteristics and spectral retention characteristics, and the overall effect is better.

In Figure 11, compared with the reference MS image, the BL and GFD methods appear spectral distortion with different degrees. In particular, in the local magnification region, for the BL method, the light gray part becomes dark gray. For the GFD method, the yellowish part becomes pink. Moreover, the WTSR and GFD methods have severe spatial detail distortion. The fused images obtained by the MOD, PRACS, VLGC, and BDSD-PC methods have a higher spectral resolution and retain the spectral information contained in the MS images. However, compared with these methods, the spatial details of the proposed method are more defined, especially in the local magnification region. In Figure 9, from the analysis of subjective vision, the proposed method has clearer spatial details and higher spectral resolution, which means that the proposed method improves spatial details while more spectral information contained in the MS images is retained.

In Table 3, the proposed method can obtain the best values on all six quantitative indices. Besides, the VLGC method can obtain the second best values on the ERGAS and QNR indices, and the third best values on the CC, SID, and RASE indices. The MOD method can obtain the second best values on the CC, SID, and RASE indices, and the third best values on the ERGAS and QNR indices. Thus, from the analysis of the objective evaluation results in Table 3, the proposed method has superior spatial detail retention characteristics and spectral retention characteristics, and the overall effect is better.

In Figure 12, compared with the reference MS image, the BL, AGS, IHST, and MPCA methods show spectral distortion with different degrees in the overall region. In particular, in the local magnification region, for the BL and IHST methods, the pink part becomes dark red and the dark green part becomes light blue. For the AGS and MPCA methods, the pink part became dark green and the dark green part became light pink. Moreover, the WTSR and GFD methods have spatial detail distortion with different degrees. The fused images obtained by the MOD, PRACS, VLGC, and BDSD-PC methods have higher spectral resolution and retained the spectral information contained in the MS images. However, compared with these methods, the spatial details of the proposed method are more defined,

especially in the local magnification region. In Figure 10, from the analysis of subjective vision, the proposed method has clearer spatial details and higher spectral resolution, which means that the proposed method improves spatial details while more spectral information contained in the MS images is retained.

In Table 4, the proposed method can obtain the best values on all six quantitative indices. In addition, the VLGC method can obtain the second best values on the ERGAS and CC indices, and the third best values on the SID, QNR and RASE indices. The MOD method can obtain the second best values on the QNR, SID and RASE indices, and the third best values on the ERGAS and CC indices. The BDSD-PC method can obtain the second best value on the SID index. The PRACS method can obtain the third best values on the CC and SID indices. The PRACS method can obtain the third best value on the CC index. Thus, from the objective evaluation results in Table 4, the proposed method has superior spatial detail retention characteristics and spectral retention characteristics, and the overall effect is better.

From the perspective of subjective vision, the fusion results obtained from the proposed method have clearer spatial details and higher spectral resolution. From the perspective of objective evaluation, the fusion result obtained from the proposed method performs best on all six quantitative indices. In general, the comprehensive evaluation based on objective evaluation and subjective visual effects can show that the proposed method has superior spatial detail retention characteristics and spectral retention characteristics.

## 5. Conclusions

In this paper, a novel MS and PAN images fusion method based on WMCF decomposition is proposed. A WLSF-based method is used to fuse all the bands of the MS image to generate the intensity component *IC*. In accordance with an image matting model, *IC* is used as the original $\alpha$ channel for spectral estimation to obtain a foreground image *F* and a background image *B*. WMCF and GF are used to decompose a PAN image into three scales, i.e., a small scale (*SS*) image, a large scale (*LS*), and a basic image, respectively. Then, a MSMDM-guided PAC-PCNN fusion strategy is used to fuse *IC* and the basic image obtained from the PAN image. All parameters in the PAC-PCNN model can be automatically calculated in accordance with the inputs with a fast convergence speed. Finally, the fused images, and the *LS* and *SS* image are combined as the last $\alpha$ channel. In accordance with an image matting model, the foreground color *F*, background color *B* and the last $\alpha$ channel are reconstructed to obtain the final fused image. The experimental results show that the method proposed in this paper has better performance than some representative pan-sharpening methods and can solve the problem of spatial distortion and spectral distortion.

We conducted experiments using four different image pairs to compare with ten representative pan-sharpening methods. The experimental results show that the proposed method can achieve superior results in terms of visual effects and objective evaluation. The proposed method can obtain more spatial details from the PAN image with higher efficiency while retaining more spectral information contained in the MS image. By fusing the MS and PAN images, the fused images have both higher spatial detail representation ability and retain the spectral features contained in the MS images, which can obtain a richer description about the target information. With the improvement of remote sensing satellite sensor technology, the information integration and processing technology has been enhanced. Thus, the proposed method can be better applied to various aspects, such as military reconnaissance, remote sensing measurements, forest protection, vegetation cover, image classification and machine vision.

In the future research, we will work on developing more effective fusion strategies and explore the application of our method in other fields.

**Author Contributions:** Conceptualization, Y.P.; methodology, Y.P.; software, S.X.; validation, S.X.; formal analysis, Y.P.; investigation, S.X.; resources, Y.P.; data curation, S.X.; writing—original draft preparation, Y.P.; writing—review and editing, Y.P., D.L. and J.A.B.; visualization, Y.P.; supervision, D.L. and J.A.B.; project administration, D.L.; funding acquisition, L.W. All authors have read and agreed to the published version of the manuscript.

**Funding:** This paper was supported by Leading Talents Project of the State Ethnic Affairs Commission and National Natural Science Foundation of China (No. 62071084).

**Institutional Review Board Statement:** Not applicable.

**Informed Consent Statement:** Not applicable.

**Data Availability Statement:** Not applicable.

**Conflicts of Interest:** The authors declare no conflict of interest.

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
