# Peer review of "A Multispectral and Panchromatic Images Fusion Method Based on Weighted Mean Curvature Filter Decomposition"

_applsci, doi:10.3390/app12178767_

Round 1

Reviewer 1 Report

The authors proposed A Multispectral and Panchromatic Images Fusion Method 2 Based on Weighted Mean Curvature Filter Decomposition, Overall, the manuscript is recommended with some revisions as stated below:

1.     Abstract
-       The Abstract section needs improvement.
-       The problem statement and the objectives of the research need to be included.
-       There are no significant contributions in the abstract showing the importance of remote sensing images.
-       There are no significant results in the abstract to show the importance of the proposed method

2. Introduction
-       I could not find any problem statement in the introduction section.
-       The authors should clearly includehe following: problem, motivation, scope, objeobjective

      - The problem statement should also address: why is it important. Why is it needed to solve?

3. Related Method
-       The related work section needs summarization, no need to write all the equations, the authors can just mention them in the text.

- Figure 1 is missing in the text (should be added)

4.Step and principles

-Please change the name of this section to Methodology or Proposed Method

- Write the steps as an algorithm

-       this section needs to be improved more, there should be some introduction and discussion on each step

5.  Experiments and Analysis

Figure 8 is missing in the text
There is no compassion with  state-of-the-art methods

Reviewer 2 Report

In the article “A Multispectral and Panchromatic Images Fusion Method Based on Weighted Mean Curvature Filter Decomposition”, the authors proposed a multi-spectral (MS) and panchromatic (PAN) image fusion method based on weighted mean curvature filter (WMCF) decomposition which focuses on solving the spatial and spectral distortion.

Given the expediency of providing a report, my comments are somewhat limited, though I hope they are still useful to the editors and authors:

The topic chosen by the authors is worthy of investigation. Overall the article is good, the methodology and procedure appear sound and the results are interesting. This paper can be considered after necessary revisions. The following issues must be addressed and clarified before acceptance of the article.

1.      The novelty of the work must be clearly mentioned in the abstract of the article as well as in the conclusion section.

2.      Include/discuss some more latest relevant studies in the introduction part.

3.      The methods part is well explained. Normally, the procedure/methodology must be comprehensive and detailed enough to be reproduced by other researchers which the authors have presented very well and their efforts are appreciated. However, if the authors think that if there some details missing which might be useful to young researchers to reproduce the results can be further added.

4.      Although the authors have compared their results well and put in the form of tables, it will further enhance the quality of this work if the quantitative results are presented in the form of plots also.

5.      I noticed several grammatical/ sentence structuring errors throughout the paper.  It is suggested to thoroughly check the manuscript and correct such type of errors.

Round 2

Reviewer 2 Report

The authors have addressed the issues/comments and incorporated the suggestions. The revisions are satisfactory and the revised manuscript is now acceptable for publication.